# Modeling of Enhanced Polar Magneto-Optic Kerr Effect by Surface Plasmons in Au Bowtie Arrays

**DOI:** 10.3390/nano13020253

**Published:** 2023-01-06

**Authors:** Jingyi Liu, Lianchun Long, Yang Yang

**Affiliations:** 1Faculty of Materials and Manufacturing, Beijing University of Technology, Beijing 100124, China; 2Beijing National Laboratory for Condensed Matter Physics, Institute of Physics, Chinese Academy of Sciences, Beijing 100190, China

**Keywords:** magneto-optical surface plasmon resonance, polar magneto-optical Kerr effect, numerical simulation, quality factor

## Abstract

The weak magneto-optical (MO) signal of traditional MO materials is indeed an important issue for their further practical applications. Although many strategies have been proposed to improve the MO effect, hybridization with noble metal nanostructures is a promising route in recent years due to the high localized-surface plasmon resonances (LSPR) effect. A new magneto-optical surface plasmon resonance (MOSPR) structure hybrid with Au bowtie arrays is proposed to increase the measuring range of the polar magneto-optical Kerr effect (PMOKE) and the quality factor through the LSPR effect. It is verified by a numerical simulation of the finite element method (FEM). The optimized parameters were found by modulating the shape and geometric dimensions. Owing to the significant LSPR from the Au bowties, a PMOKE amplification signal spectrum with narrow linewidth, and a high amplitude with high-sensing performance was achieved. Compared with the bare magnetic film alone, by optimizing the relevant parameters of the LSPR structure, the maximum signal increases 3255 times, and the quality factor can be greatly improved, which would provide important guidance and help for the practical application of MO devices.

## 1. Introduction

Due to the wide range of applications of magneto-optical (MO) devices, effective characterization methods for MO materials are needed to explore new MO materials and enhance the performance of currently used ones [1]. By detecting the interaction between light and MO materials, the MO characterization technology based on the magneto-optical Kerr effect (MOKE) has achieved in situ nondestructive detection [2,3,4]. Based on the relative orientation between the magnetic field and the plane of incidence, the MO effects in reflection can be categorized into three types: polar (where the magnetic field is perpendicular to the reflection surface and parallel to the plane of incidence), longitudinal (where the magnetic field is parallel to both the reflection surface and the plane of incidence), and transverse (where the magnetic field is perpendicular to the plane of incidence and parallel to the reflection surface) Kerr effects. The MO rotation of polar and longitudinal magneto-optical Kerr effects is proportional to the magnetization. Generally, the polar effect is the strongest, followed by the longitudinal effect, and there is no obvious MO rotation in the transverse effect [5]. Based on this, relevant research has established a detection method for magnetic nanomaterials [6,7,8].

Normally, the MO effect of traditional materials is relatively weak and almost unmodulable, thus the Kerr rotation angle and Kerr ellipticity are very small. It is difficult to achieve accurate measurements by conventional measurement methods [9], which limits its application in devices. Therefore, improving the observation accuracy of magneto-optical Kerr is still an important issue [10]. Magneto-optic surface plasmon resonance (MOSPR) can be controlled by external magnetic fields, overcoming this limitation [11,12]. Early MOSPR investigations concentrated on a few flat, pure ferromagnetic films, and to improve the MO effects, surface plasmon resonance (SPR) was activated on the surface of these films [13,14,15]. Later, ferromagnetic materials and noble metals were combined to create excellent magnetic plasmon devices that benefited from both the plasmon properties of noble metals and the MO properties of ferromagnetic materials [16,17,18]. Due to the great LSPR, the MOKE in the noble metal–ferromagnetic system can be significantly increased compared to the pure ferromagnetic material [19,20]. Based on this, researchers have looked into the MO effects of multilayer film structures with holes on noble metals [21,22]. Significant changes in the MO effect can be seen by etching grating strips [23,24], nanodot arrays [25,26], nanopore arrays [27] and rectangular apertures, nano-pillars, C-apertures/C-engravings, etc. [28,29], onto noble metal–ferromagnetic–noble metal films. The existing results show that the bowtie plasmon structure enhances the electric field strength and the absorption [30]. At the same time, the results show that the optical response is related to the side length of the structures. The bowtie structure is selected as the plasmon structure in this paper [31]. With the deepening of research, it is still a major challenge to design controllable systems with excellent plasmon and MO characteristics. It is an effective way to optimize the design of MOSPR structures from many aspects such as material, structure, and size [32]. Some studies have obtained the quality factor and monitoring limit beyond LSPR devices based on the MOSPR structure. For example, MacCaferri et al. [33] created a short-range ordered nickel disk-shaped nanoantenna structure utilizing glass as a substrate. However, the existing devices’ high losses and low-quality factor, together with the inherent scattering losses, make the PMOKE spectrum very wide, which still restricts the improvement in device excellence. Therefore, the realization of PMOKE with narrow linewidth and high intensity through structural design is an important direction for the research of new high-quality-factor MOSPR sensor devices.

In this paper, a finite element model to simulate the MO effect has been developed. We determined the Kerr rotation and ellipticity of the Au nanostructure–permalloy hybrid structure by integrating the semi-analytical approach with the FEM, and the quality factor was added to quantitatively evaluate the improvement in the sensor performance. The influence of the distribution form, side length, and period are studied. By optimizing the geometry parameters of Au bowties, this paper demonstrates that the maximum PMOKE signal can be promoted up to 3255 times that of magnetic films alone, and that its quality factor is significantly increased, which will be used extensively in upcoming MO devices.

## 2. Computational Methods

The complexity of the interaction of polarized light with magnetic materials under magnetic field loading makes traditional analytical methods infeasible, especially for structures with complex boundary conditions.

Using experimental means to accurately measure the MO Kerr rotation angle and realize MOKE enhancement to screen the best parameters is time-consuming and costly, but the numerical simulation method can effectively address this issue. Permalloy is frequently employed in mild magnetic fields in sensor systems with stringent sensitivity requirements because of its high permeability, zero magnetocrystalline anisotropy, low coercivity, and microwave damp [34,35,36]. Au nanostructures exhibit excellent plasmonic characteristic [37,38], which could effectively enhance the optical properties of devices. For structures with complex boundary conditions, it is impossible to solve the magneto-optical coupling problem analytically. Therefore, it is necessary to carry out a simulation calculation through a numerical simulation, and any complex structure can be modeled and analyzed by using a suitable finite-element modeling program. In this paper, COMSOL Multiphysics is used for calculation, in which the dielectric constant is defined as a complex non-Hermitian tensor to better describe the MO effect [39]. The total dielectric constant of the magnetic layer is defined as the complex tensor, as shown in Equation (1).
(1)$$\varepsilon_{\mathrm{MO}}=\varepsilon_{0}\varepsilon_{r}\left[\begin{array}{ccc} 1 & iB_{z}Q & -iB_{y}Q\\ -iB_{z}Q & 1 & iB_{x}Q\\ iB_{y}Q & -iB_{x}Q & 1 \end{array}\right]$$
where *B_i_* is the cosine of the direction of magnetization along the *i*-axis, and *Q* is the complex Voigt constant (the property of a specific magnetic material). The first-order reflection coefficient of the interface between a non-magnetic medium with refractive index *n*_0_ and a magnetic medium with refractive index n1 is defined as [40,41]:(2a)$$r_{ss}=\frac{n_{0}\cos\theta_{0}-n_{1}\cos\theta_{1}}{n_{0}\cos\theta_{0}+n_{1}\cos\theta_{1}}$$
(2b)$$r_{pp}=\frac{n_{1}\cos\theta_{0}-n_{0}\cos\theta_{1}}{n_{1}\cos\theta_{0}+n_{0}\cos\theta_{1}}+\frac{2iQn_{1}n_{0}\cos\theta_{0}\sin\theta_{1}B_{y}}{{\left(n_{1}\cos\theta_{0}+n_{0}\cos\theta_{1}\right)}^{2}}$$
(2c)$$r_{ps}=\frac{iQn_{0}n_{1}\cos\theta_{0}(\sin\theta_{1}B_{x}-\cos\theta_{1}B_{z})}{\cos\theta_{1}(n_{0}\cos\theta_{0}+n_{1}\cos\theta_{1})(n_{1}\cos\theta_{0}+n_{0}\cos\theta_{1})}$$
(2d)$$r_{sp}=\frac{iQn_{0}n_{1}\cos\theta_{0}(\sin\theta_{1}B_{x}+\cos\theta_{1}B_{z})}{\cos\theta_{1}(n_{0}\cos\theta_{0}+n_{1}\cos\theta_{1})(n_{1}\cos\theta_{0}+n_{0}\cos\theta_{1})}$$
where *r_pp_* and *r_ss_* are Fresnel reflection coefficients, and *r_ps_* and *r_sp_* are MO reflection coefficients, *θ*_0_ and *θ*_1_ are the complex refraction angles in a non-magnetic medium and a magnetic medium, respectively. The analysis calculation assumes that the change in field quantity in space and time is exp(*i*(*k·r*) − *ωt*)), where *k* is the wave vector, *r* is the position vector, *ω* is the angular frequency, and *t* is the time. This form requires the imaginary part of the refractive index to be positive. The polarization rotation *ϕ_p_* and the ellipsoidal deflection *ξ_p_* are
(3)ϕp=−Re{rsprpp}ξp=Im{rsprpp}

Combined with Equations (2a–d) and (3), for the magnetic film alone without Au hybridization, when the magnetization is in a polar configuration, the analytical expression of semi-infinite thick-plate materials predicts that the vertical-incidence Kerr rotation and ellipticity are 74.7 mDeg and −24.7 mDeg, respectively. When Au hybridization is introduced into the structure, the PMOKE response cannot be calculated analytically. Therefore, we need to use the FEM to calculate the PMOKE response of the structure.

The incident plane wave in the calculation model is parallel to the plane magnetosphere’s surface, and periodic boundary conditions reduce the model’s geometric structure while maintaining accuracy. Therefore, the calculation model represents an infinite thin film, and only the representative volume element is displayed. To prevent diffraction modes, the chosen Au model’s period needs to be smaller than the incident wavelength’s vacuum wavelength. Four possible symbols are generated when describing MO permittivity because two self-consistent definitions of complex permittivity are generated by choosing a symbol convention when describing wave propagation [42]. In the setting of material properties, permalloy’s refractive index *n* = 2.25 + 3.7*i*, Voigt constant *Q* = −0.006 − 0.011*i*, and dielectric constant were obtained from reference [43] *ε*_mag_ = −8.628 + 16.65*i*. They are wavelength-independent fixed values, so all wavelength-dependent responses are caused by Au rather than permalloy magnetic films. The reflected wave generated by the rotation and ellipticity caused by the PMOKE has an electric field component in the Y direction. Complex field components are extracted by a certain plane. The Kerr rotation *ϕ* and ellipticity *ξ* are calculated by Equation (4) [44].
(4)ϕ=(ReEx)(ReEy)+(ImEx)(ImEy)(ReEx)2+(ImEy)2ξ=(ReEx)(ImEy)−(ImEx)(ReEy)(ReEx)2+(ImEy)2

This expression applies to the non-transparent semi-infinite films with small *ϕ* and *ξ* values. Adjust the mesh division of the model to avoid the inconsistency of the Kerr rotation on different reflection planes. It is computationally impossible to solve the field in an infinite region. Therefore, various strategies can be used to truncate the model to a reasonable size. The perfect match layer (PML) belongs to this truncation strategy. The PML domain function is used for stationary control equations that are essentially waveforms, acting as an almost ideal absorber or radiator domain, simulating the assumption of an infinite thin film. Mathematically, PML is a domain with anisotropy and complex permittivity and permeability. Although there is no reflection in theory, due to the numerical dispersion (mesh), there will still be some reflection. To minimize this reflection, a mesh consistent with anisotropy in material properties is used in PML. In addition, the interface between the air domain and the magnetic material is refined locally to ensure a stable Kerr signal. To verify the reliability of the FEM calculation, taking the polar-magnetized magnetic-film-alone structure as the initial structure, the PMOKE rotation and ellipticity on the detector plane are calculated. The model-size and boundary-condition settings are the same as those in Figure 1. Initial polar MOKE modeling produced a Kerr rotation of 75 mDeg and an ellipticity of −25 mDeg, with a variation between different models of ≈1 mDeg. These values are consistent with the values of 74.7 mDeg and −24.7 mDeg calculated from analytical equations.

## 3. Result and Discussion

### 3.1. Modeling and Verification

To increase the PMOKE, a Au bowtie structure was designed, and the effect of the geometric parameters on the MOSPR structure’s Kerr signal was investigated. The calculation model is under normal incidence, and the representative volume element with the Period X * Period Y section in the infinite array film is taken for analysis, as shown in Figure 1. The distance between the two tips of the bowties with side length *a* and height H is *δ*, and the relative rotation is *θ*. The centroid of the two Au bowties is always horizontal, so any one of the three variables Period X, *δ* and *a* can be represented by the other two; that is, they are dependent variables. In this paper, Period X = Period Y = 400 nm is maintained in all models. The MOSPR device considered in this paper has a special direction on the surface, which can be represented by the bowtie-shaped axis of symmetry. The MO coupling response depends on the angle between the polarization direction and this axis of symmetry. The polarization direction of the incident light in this model is consistent with this axis, so this effect is ignored.

In order to verify the modulation advantage of the structure on the PMOKE, the Kerr rotation and the ellipticity of equal volume Au bow and Au nanodisk are calculated with the incident wavelength and side length *a* as variables, as shown in Appendix A.

### 3.2. Effects of Geometric Parameters of Nanobowties

#### 3.2.1. Height

Firstly, the influence of height H on the Au nanostructure–permalloy hybrid structure’s PMOKE is studied. Keeping the side length *a* equal to 100 nm, the tip of the bowties has no relative rotation and 10 nm distance. Calculate the change in the PMOKE when H changes from 20 nm to 100 nm. The results are shown in Figure 2a. It can be seen that, as the height increases, the Kerr rotation has a peak at the resonance wavelength. The Kerr rotation increases dramatically when the height exceeds 40 nm. The electric field distribution cloud map reveals that, at the resonance wavelength, the electric field stimulates LSPR near the Au bowties, causing the strength in the immediate area to grow greatly, and hence the Kerr rotation to increase significantly. A finer exploration of the near-field region of the bowtie center was performed. The X-direction electric field component at the center of the Au nano bowties was extracted, and Figure 2b shows its relationship with the wavelength. When the resonance condition of gold nano-particles is satisfied, an evident enhancement of the MO activity is predicted to exist. An increased MO activity appears at the same spectral position of the increased electric field associated to the LSPR excitation, suggesting an intimate connection between plasmonic and MO effects. Height changes only influence the PMOKE rotation value, not its regularity with wavelength change or resonant wavelength. Since height is not the main focus of this article, we have adopted a set height to prevent any changes that might result from a change in height without losing its generality. Each representative volume element contains Au bowties with a height of 50 nm.

#### 3.2.2. Gap Distance

The influence of gap *δ* on the MOKE is investigated further. It has a fixed side length of 100 nm, a height of 50 nm, no relative rotation, and *δ* ranging from 0 to 80 nm, with 80 nm being the largest permitted separation size in the 400 × 400 nm^2^ sector. Due to particle form and size, the region described here is outside the effective range of the simplified Mie scattering theory and the dipole approximation. As a result, the complete analytic solution is outside the scope of this work. The absolute Kerr signal representation of Kerr rotation and the ellipticity are summed in orthogonal form, since the interplay between rotation and ellipticity is complicated. Figure 3 depicts the changing rule for the Kerr rotation, ellipticity, and absolute Kerr signal with *δ*. It is not difficult to find that *δ* has little influence on the PMOKE, and the absolute Kerr signal is represented as an index. As a result, it will no longer be considered a variable in the next calculation. The fixed *δ* = 10 nm is used in the following calculation model to explore the influence of various geometric parameters.

#### 3.2.3. Side Length

The influence of side length a on the PMOKE of the whole structure is studied under the condition that height H and gap *δ* are unchanged. Side length *a* varies from 100 nm to 230 nm. Modeling geometry allows for a maximum side length of 230 nm. The relationship between PMOKE and wavelength under different side lengths is shown in Figure 4. It is not difficult to find that changing the side length will change the resonance wavelength of the structure. Figure 4a,b indicate that the structure enhances the PMOKE signal at resonance regardless of the side length. Within the side length of 100 nm–230 nm, the maximum and the minimum resonance signals are, respectively, 1434.29 times and 19.90 times the absolute PMOKE signal of the magnetic film alone without Au nanostructure. The increase is due to the enhancement of interaction between the electric field and the MO active material due to the change in the electric field near the array structure during resonance. The signal gradually strengthens as the side length increases, reaching a maximum when the side length is 150 nm. Later, as the side length increases, the PMOKE exhibits a diminishing oscillation tendency. The PMOKE shows an oscillatory downward trend when the side length increases. The general trend of the Kerr signal at resonance is due to the change in the interaction area between the enhanced near-field and the magnetosphere. In all structures, the form of the resonance curve follows a similar pattern, because the two components of the PMOKE (rotation and ellipticity) show similar bimodal modes. With 150 nm as the dividing line, the structures exhibit a more superior low-frequency mode MOKE with increasing side length, and this effect exists in the regulation of Kerr rotation, ellipticity, and absolute PMOKE signal, but has no effect on the structure’s resonant wavelength; that is, as side length increases in the ranges of 100–150 nm and 160–230 nm, the resonance of the structure gradually shifts from high-frequency mode to low-frequency mode. Therefore, the quantitative regulation of the signal can be achieved by reasonably selecting the side length.

LSPR activation is carried out by adjusting the wavelength of the incident light. When the resonance adjustment of the Au nano-bowties structure is satisfied, the activity of the PMOKE is significantly enhanced, as shown in Figure 4a. As shown in Figure 4c,d, when LSPR is activated, a significant enhancement of the electric field can be observed near the particle surface. According to reference [38], when the resonance condition of gold nano-particles is satisfied, an evident enhancement of the MO activity is predicted to exist. An increased MO activity appears at the same spectral position of the reflectivity reduction associated to the LSPR excitation, suggesting an intimate connection between plasmonic and MO effects. Similarly, the LSPR excited at the Au nano-bowties structure causes local enhancement of the surrounding electric field and significant enhancement of the absolute PMOKE signal. In addition, the 100 nm side length shows a relatively symmetrical field around the bowties, while for the 220 nm side length, the field is concentrated on the side of the bowties facing the magnetic material. At the optimal side length, the response of the Kerr signal representation is significantly increased in comparison to that of the magnetic film alone. This enhancement represents a significant increase in the PMOKE signal predicted by the system for unchanged magnetic materials. The electric field intensity of the structure increases in both the X and Y directions due to the action of the plasmons. According to the simulation results, changing the size of the structure will result in a change in the electric field intensity in both directions, and this change rule will have a maximum. As a result, at a specific side length, the structure will maximize the magneto-optical coupling effect. At that point, the electric field intensity in the X direction is 43% higher than that of the pure magnetic film structure, while it is 72% higher in the Y direction, resulting in the maximum absolute Kerr signal.

#### 3.2.4. Relative Rotation

Apart from *δ* and *a*, a rotation variable *θ* is introduced. According to the above results, the side length of the Au bowties has a more significant effect on the PMOKE than the gap distance. Therefore, the response with five side lengths of 100 nm, 125 nm, 150 nm, 175 nm, and 200 nm as a function of *θ* and wavelength are calculated, respectively, as shown in Figure 5. The PMOKE changes with *θ*, and this change is closely related to the side length. It can be seen in Figure 5a,c,e,g,i that the change in side length has little effect on the resonance wavelength. To facilitate comparison, the curves of absolute Kerr signals with different side lengths at resonance wavelengths 565 nm, 665 nm, and 815 nm versus the rotation are drawn, as shown in Figure 5b,d,f,h,j.

It can be seen in Figure 5 that increasing *θ* results in a signal augmentation for the PMOKE with the same side length. The maximal signal is greater than that of the structure without the rotation angle. The rotation angle does not affect the structure’s resonant wavelength. The structure’s resonant signal achieves its maximum value when *a* = 150 nm, *θ* = 19° (Figure 5f, red line), which is 2.3 times stronger than the structure without rotation angle at the same side length, and 3255 times stronger than the magnetic film alone. Comparing the structures with different side lengths, the maximum signal shifts from multi-resonant frequency mode to single-resonant frequency mode. The disparity between signals at different angles grows significantly as the side length increases. The resonance frequency corresponding to the maximum value of the PMOKE shifts from high to low frequency, and the structural angle corresponding to the maximum value shifts from large to small rotation angle. In other words, when the structure’s side length exceeds 150 nm, the PMOKE shifts from multipole to monopole mode with angle change, making it easier to alter the signal by adjusting the structure angle.

In order to deeply understand the spectra generated by different values of *θ*, Figure 6 shows the comparison of electric field distribution around bowties when the side length is 175 nm and *θ* changes from 1° to 17° under a 565 nm incident wavelength. Since the side length and angle change at the same time, the resonance wavelength of the structure will be affected, that is, the resonance wavelength of the maximum PMOKE of different structures is not exactly the same. It can be seen in Figure 6 that the X component of the electric field around the bowties is locally enhanced at the two tips of the upper edge of the structure. With the increase in rotation angle, this local enhancement effect transits from unilateral enhancement of the left tip to bilateral enhancement of the left and right tips, and finally to unilateral enhancement of the right tip. When the rotation angle is 9°, the effect of the local electric field around the structure to enhance the penetration into the air reflection area is most obvious, so the PMOKE is most significant here. The results are shown in the black curve in Figure 5h.

Obviously, from the results shown in Figure 2, Figure 3, Figure 4, Figure 5 and Figure 6, the observed behavior is very sensitive to the bowties’ side length and rotation. The joint change in side length and angle leads to a change in the area of the incident wave acting on the bowties’ structure, resulting in a different local electric field distribution. The two parameter changes obtained from the research results determine two important relationships, which can guide the design of MOSPR structures. Bowtie geometry has a great influence on the resonance wavelength and a certain influence on the resonance enhancement. It has the best combination of side length and rotation. By manipulating these two basic parameters, it should be possible to tune the MOSPR structure to the desired wavelength, and then maximize the observed Kerr signal by bringing it sufficiently close to the magnetic material.

### 3.3. Quality Factor

A more representative index is needed to measure the enhancement of the PMOKE to better capture the influence of geometric parameter changes. The plasmon nanoarray structure is typically employed for sensor detection, and the quality factor is utilized to assess its performance. Here, we define it as the proportion of the absolute Kerr signal to the peak width of the spectrum (half linewidth of the resonant peak). When *a* is 150 nm and *θ* is 19°, the quality factor of the newly designed MOSPR structure reaches its maximum, about 51,461, which is much higher than that reported in previous studies [45,46,47,48], as shown in Figure 7. The MOSPR detector obtained a crisp PMOKE spectrum with high amplitude and narrow bandwidth by adjusting the *a*, *δ*, and *θ* of Au bowties. Because of the high-quality LSPR effect, detection performance has been greatly increased, allowing for a broader range and an improved signal.

## 4. Conclusions

In this paper, a new Au nanostructure–permalloy hybrid structure is designed, and the wavelength-dependent MOKE resonance signal is obtained, which confirms the coupling enhancement effect of local-surface plasmon resonance on the polar magneto-optical Kerr effect. The signal intensity and the detection range of the polar Kerr effect increase in these structures. The signal intensity at resonance is primarily determined by the side length and the relative rotation angle of the Au bowties. The results show that, compared to films without Au nanostructures, the MOSPR structure produces a signal enhancement of the polar magneto-optical Kerr effect of up to over 3255 times. By selecting the appropriate geometric size and layout of the Au bowties, the signal can be amplified in the visible and near-infrared shortwave ranges. The quality factor of this enhanced structure is significantly higher than that of the traditional nano-composite magneto-optical film structure, promising its potential application in magneto-optical devices. The FEM framework of the PMOKE signals used to simulate MOSPR structures in the whole region is constructed by reasonably setting the geometry and boundary conditions of the model, which allows one to model plasmonic resonators of more complicated geometry and composition, and their interaction with magneto-optically active materials. The model is established based on certain ideal assumptions and has certain limitations. Its results can provide a guiding idea for experimental testing and new structure design, but they cannot replace the experiment completely. This work will affect the future of detector design and may lead to more sensitive or high-resolution magneto-optical imaging detectors.

## Figures and Tables

**Figure 1 nanomaterials-13-00253-f001:**
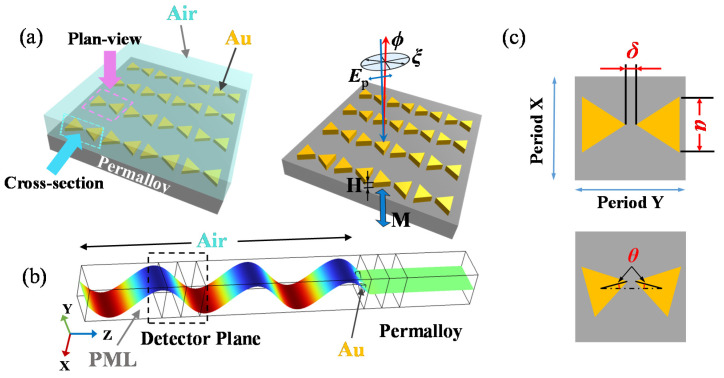
Schematic of the periodic model geometry for calculation of the PMOKE. (**a**) General schematic showing the Au nanostructure, magnetic region, air region, and p-polarized light incidence mode (the polarized light is incident vertically, and the polarization angle is 0°). (**b**) The parameters that affect the PMOKE discussed in this paper. (**c**) The X directional component of the electric field, which is the result of the MO rotation induced by the PMOKE.

**Figure 2 nanomaterials-13-00253-f002:**
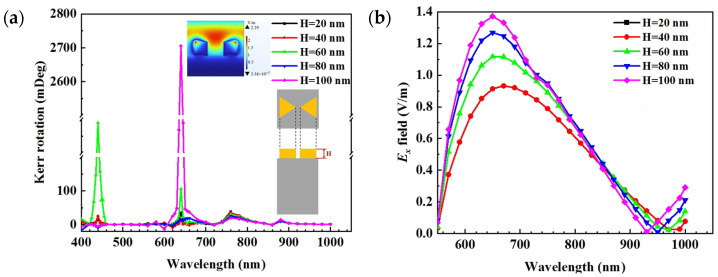
Wavelength-dependent PMOKE response for Au bowties of different height (heights are indicated in the legend). The bowties were located centered within the unit cell of the periodic model. The color subgraph represents the reflected electric field. (**a**) Kerr rotation and (**b**) normalized amplitude of the X component of the reflected electric field.

**Figure 3 nanomaterials-13-00253-f003:**
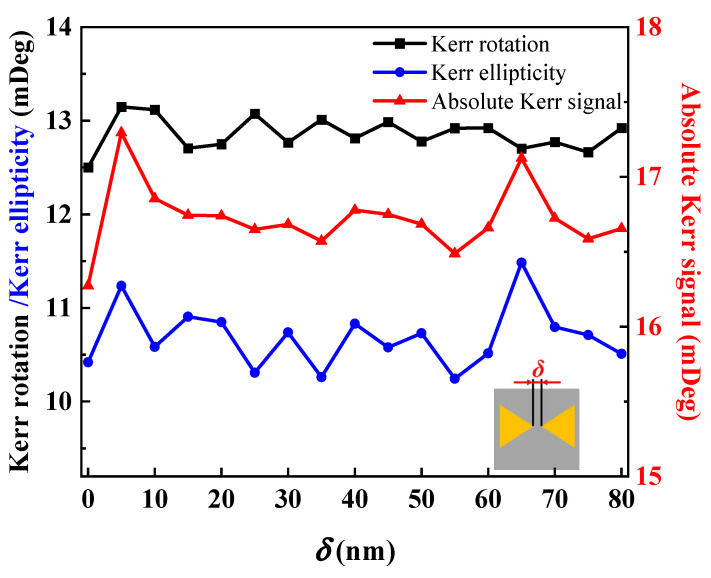
Kerr rotation, Kerr ellipticity, and absolute Kerr signal (the sum of the rotation and ellipticity in quadrature) of PMOKE response for Au bowties of different gap distance *δ*.

**Figure 4 nanomaterials-13-00253-f004:**
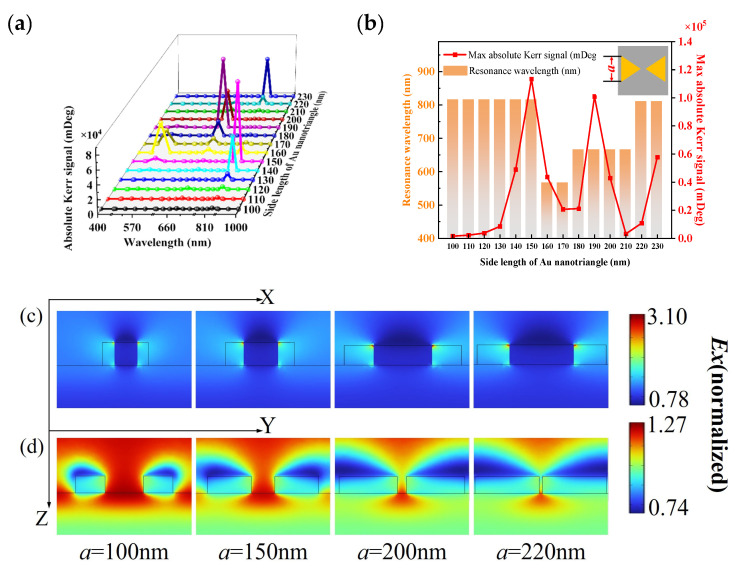
Wavelength dependence of PMOKE response for the Au bowties of different side length. (**a**) shows the absolute Kerr signal with difference in side length from 100 nm to 230 nm, (**b**) the maximum absolute Kerr signal and resonant wavelength, (**c**) the normalized amplitude of the X component (*Ex*) of the electric field in the XZ plane, and (**d**) the normalized amplitude of the X component (*Ex*) of the electric field in the YZ plane.

**Figure 5 nanomaterials-13-00253-f005:**
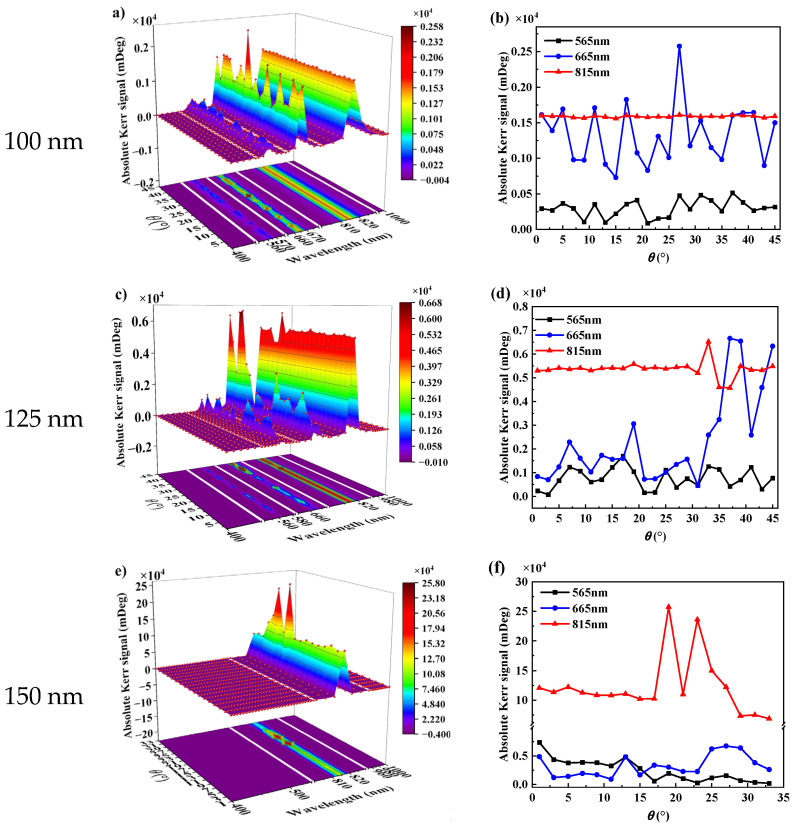
Wavelength dependence of PMOKE response for the Au bowties of different *θ* and side length placed at center of the magnetic layer (the side length is indicated in the legend). (**a**,**c**,**e**,**g**,**i**) indicate the change rule in PMOKE with *θ* under 400–1000 nm incident wave when the side length changes from 100 nm to 200 nm every 25 nm. (**b**,**d**,**f**,**h**,**j**) denote the change rule in PMOKE of the structure with *θ* under three resonant wavelengths (565 nm, 665 nm, and 815 nm).

**Figure 6 nanomaterials-13-00253-f006:**
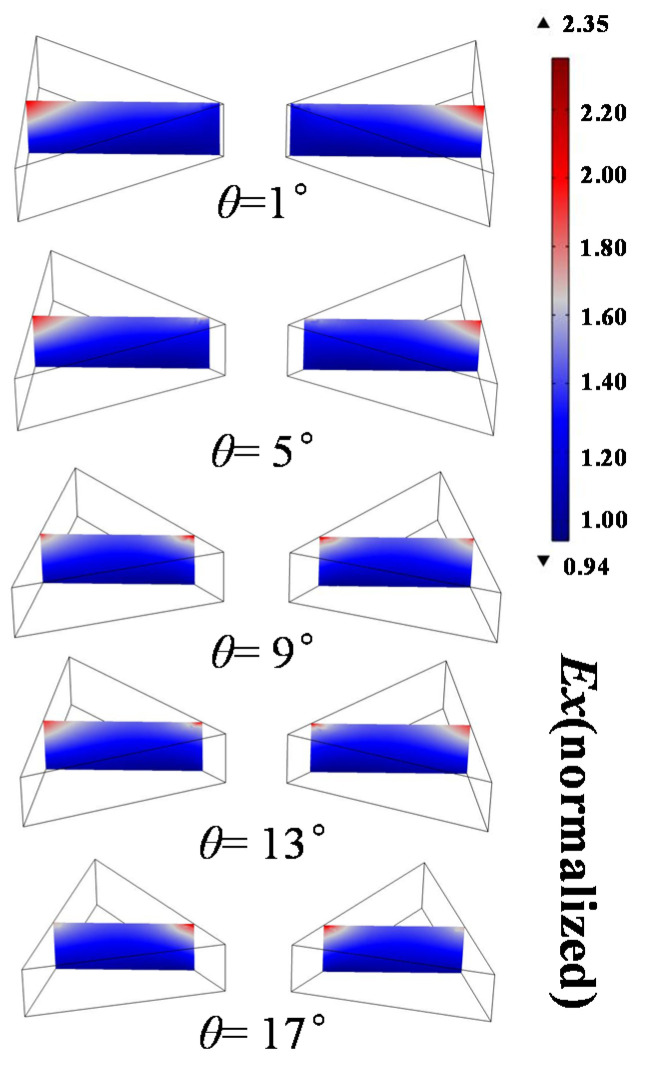
Electric field distribution of 175 nm side length Au bowties placed at rotations of 1°, 5°, 9°, 13°, and 17° under 565 nm incident wavelength. The cross-sections show the X component of the local field, normalized by the incident field amplitude, plotted at one instance in phase. Field distribution for a cross-section through the center of the YZ plane. The field appears to penetrate further into the reflect air layer in the case of a 9° rotation.

**Figure 7 nanomaterials-13-00253-f007:**
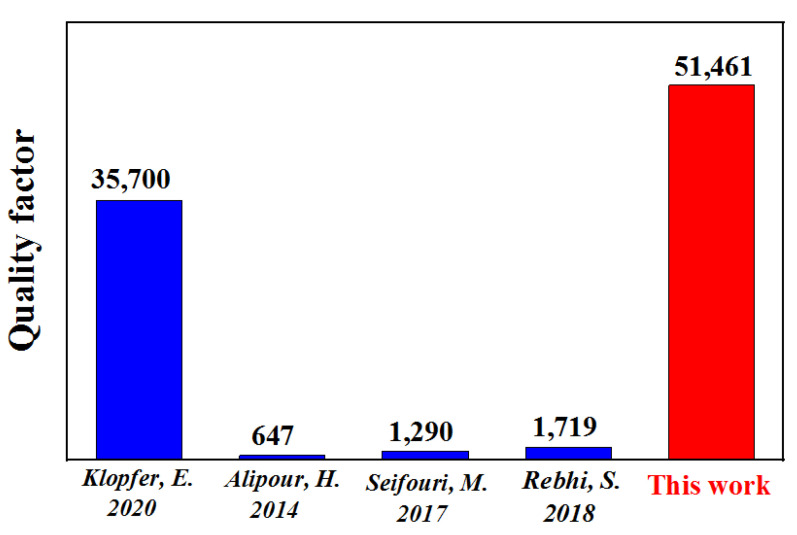
Comparison of quality factors amongst different structures. (Klopfer, E. 2020 [45]; Alipour-Banaei, H. 2014 [46]; Seifouri, M. 2017 [47]; Rebhi, S. 2018 [48]).

## Data Availability

No data were used for the research described in the article.

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
