# Peer review of "Modeling of Enhanced Polar Magneto-Optic Kerr Effect by Surface Plasmons in Au Bowtie Arrays"

_nanomaterials, 2023, doi:10.3390/nano13020253_

Round 1

Reviewer 1 Report

Increasing the sensitivity of the magnetic response of the system to the effects of photons using plasmon resonance is an urgent problem, therefore the reviewed article is of interest.

The paper proposes the Au nano/permalloy structure and performs only numerical simulation of PMOKE depending on the parameters of Au nanoparticles without experimental studies.

1) Therefore, the paper should be called "Modeling of  Enhanced Polar Magneto-optic Kerr Effect by Surface Plasmons  in Au Bowties Arrays".

2) For the same reason,  in the Figure 7. Comparison of quality factors among different structure,

 it is necessary to leave only the works in which the simulation results are also presented. Because it is obvious that the experimental results will always be very different from the calculations.

3) In addition, the novelty of the results obtained should be emphasized in the Conclusion and in the Abstract. What is it: is a new structure proposed or is it proposed to use plasmon resonance in PMOKE for the first time?

Reviewer 2 Report

Comments and Recomendations:

 1. All formulas in the article are numbered. But all of them are not mentioned in the text, for example. It's not clear why formulas should be numbered if they are not referenced in the article.

2. The authors obtained their results using numerical calculations using a ready-made computer program that produces the final result without explaining what caused it. The slightest failure can lead to its distortion. This can only be understood if the latter is strongly contrary to common sense. Therefore, it is very important to give some simple estimates based on an understanding of the physical essence of the phenomenon under discussion. This would significantly strengthen the thoroughness of the work and essentially increase the credibility of the results presented.

3. The paper proposes a new hybrid of the structure of magneto-optical surface plasmonic resonance with arrays of Au bowties to amplify the magneto-optical signal. What possible ways to create such a structure do the authors see in real conditions? How feasible are all those adjustments that are easily made to the mathematical model?

4. The authors wrote about the practical significance of their article. But what is the scientific achievement of the work done? What scientific results of this work could be useful to other scientists in their research activities? This should be mentioned at least briefly, for example, in the conclusion.

This paper is well enough written to understand main results. The manuscript seems to be suitable for publication. I incline to recommend it for publication in the Journal of Nanomaterials after minor mentioned corrections.

Reviewer 3 Report

The paper proposes a gold bowtie arrays to generate surface plasmons that enhance Kerr effect. The work is interesting and should of interest to the readers of the journal. I recommend the following minor modifications before the paper can be accepted for publication.

1. The resolution of Fig. 2 is not sufficient. The peaks are not smooth. Please perform the wavelength sweep at small increments so that the curve is smoother.

2. Please define MO inside the main text even though it has been defined in the abstract. This is common practice.

3. Fig. 4b does not seem to show any regular relationship between the side-length parameter and the resonance wavelength. Please comment.

4. The optical response of plasmonic bowtie structures are dependent on the polarization of incident light. Please comment on how these effects the performance of the device. Also mention for what polarization angles were used to generate the figures at the caption of the figures.

5. It might be interesting to relate the electric field enhancement at the near-field region wit the observed Kerr rotation. Please plot a separate figure showing the electric field intensity near the bowtie center vs wavelength and relate it to the observations of Fig. 2.

6. The manuscript does not discuss in sufficient detail why the bowtie structure was used. There are other plasmonic structures like rectangular apertures, nano-pillars, C-apertures/C-engravings etc. In addition to mentioning the advantage of bowties, the authors may choose to cite a few papers that discuss other plasmonic structures such as the following:

a. https://doi.org/10.1007/s11468-022-01735-3

b. https://doi.org/10.1002/smll.202007222

7. The work is purely simulation based and has not been experimentally verified. The authors should mention some limitations of the study and the difficulties of implementing the structure in practice.
